# Preparation and Characteristics of Wet-Spun Filament Made of Cellulose Nanofibrils with Different Chemical Compositions

**DOI:** 10.3390/polym12040949

**Published:** 2020-04-19

**Authors:** Chan-Woo Park, Ji-Soo Park, Song-Yi Han, Eun-Ah Lee, Gu-Joong Kwon, Young-Ho Seo, Jae-Gyoung Gwon, Sun-Young Lee, Seung-Hwan Lee

**Affiliations:** 1College of Forest & Environmental Science, Kangwon National University, Chuncheon 24341, Korea; chanwoo8973@kangwon.ac.kr (C.-W.P.); pojs04@kangwon.ac.kr (J.-S.P.); songyi618@kangwon.ac.kr (S.-Y.H.); laa3158@kangwon.ac.kr (E.-A.L.); 2Kangwon Institute of Inclusive Technology, Kangwon National University, Chuncheon 24341, Korea; gjkwon@kangwon.ac.kr; 3Department of Advanced Mechanical Engineering, Kangwon National University, Chuncheon 24341, Korea; mems@kangwon.ac.kr; 4National Institute of Forest Science, Seoul 02455, Korea; gwonjg@korea.kr (J.-G.G.); nararawood@korea.kr (S.-Y.L.)

**Keywords:** cellulose nanofibril, wet-spun fiber, filament

## Abstract

In this study, wet-spun filaments were prepared using lignocellulose nanofibril (LCNF), with 6.0% and 13.0% of hemicellulose and lignin, respectively, holocellulose nanofibril (HCNF), with 37% hemicellulose, and nearly purified-cellulose nanofibril (NP-CNF) through wet-disk milling followed by high-pressure homogenization. The diameter was observed to increase in the order of NP-CNF ≤ HCNF < LCNF. The removal of lignin improved the defibrillation efficiency, thus increasing the specific surface area and filtration time. All samples showed the typical X-ray diffraction pattern of cellulose I. The orientation of CNFs in the wet-spun filaments was observed to increase at a low concentration of CNF suspensions and high spinning rate. The increase in the CNF orientation improved the tensile strength and elastic modulus of the wet-spun filaments. The tensile strength of the wet-spun filaments decreased in the order of HCNF > NP-CNF > LCNF.

## 1. Introduction

Cellulose is known as the most abundant biopolymer and a representative of eco-friendly materials [1,2]. It is mainly found in plant cell walls in the form of fibrils [3,4,5] and has several advantages such as low cost, low density, biodegradability, sustainability, being nonhazardous, and excellent mechanical properties [5,6]. Thus, it demonstrates great potential for application in various areas, namely, as a functional fiber and reinforced composite as well as in medical material and tissue engineering [1,7,8,9,10]. In textile industries, cellulose has been used as long filament, commonly produced through the wet-spinning process of dissolving the cellulose in solvents [11,12,13,14]. Viscose and Lyocell rayon, as regenerated cellulosic fibers, are known to be the most representative commercially manufactured fibers, which are produced from natural sources. To produce Viscose rayon, 16–19% of sodium hydroxide treated cellulose is first treated with carbon disulfide to generate sodium cellulose xanthate. The Viscose rayon is then generated via the wet-spinning of the dissolved xanthate in a 2–5% sodium hydroxide solution into a sulfuric acid solution. For Lyocell rayon production, *N*-methylmorpholine *N*-oxide has been mostly used to dissolve the cellulose, and afterwards, dry-jet-wet spinning process is carried out in a water bath to produce the filament [11,13,14,15,16]. However, these manufacturing processes have been found to be extremely harmful to the environment and require more energy, especially while recovering the used solvents [11,16]. 

Mostly, natural cellulosic fibers have been found to originally have short lengths of 10–65 mm and diameters in microns. To produce a long filament from natural cellulosic fiber, it must be spun by yarning. However, due to weak bonding between the fibers, the strength of the natural yarn fiber is lower than that of the regenerated cellulosic fiber. Thus, the fabrication of a high-strength long filament from natural cellulose fiber, especially from cellulose nanofibril (CNF) [4,8,17,18,19,20], is challenging. CNF can be generally isolated from plant cell walls by a mechanical defibrillation process, such as wet disk-milling (WDM), high-pressure homogenizing (HPH), and ball-milling [3,6,21,22,23]. The CNF has been observed to be approximately 10–30 nm in diameter, with excellent properties such as large specific surface area, good thermal properties, and high strength [6,9,21,22]. The CNF is not soluble in water but can be highly dispersible with high viscosity in water [4,10,17]. Due to these properties, CNF has been considered suitable for wet spinning to prepare a long filament [4]. Iwamoto et al. (2011) prepared the wet-spun filament from hydrogel of 2,2,6,6-tetramethylpiperidinyl-1-oxyl (TEMPO)-oxidized CNFs (TOCNF) from wood and tunicate [4]. They found the tensile strength and elastic modulus to be 90–406 MPa and 8.4–23.6 GPa, respectively, based on the CNF type and spinning rate. 

Lignocellulosic nanofibril (LCNF), containing both hemicellulose and lignin, can be prepared through mechanical defibrillation, and its surface properties can be adjusted by controlling the contents of both the components [21,22,24]. The surface of LCNF has been noted to be more hydrophobic, due to lignin. Thus it has better dispersibility in hydrophobic polymers [21,25]. Holocellulose nanofibril (HCNF) can also be prepared by removing only lignin and has been observed to have a core-shell structure, in which hemicellulose has been found to cover the cellulose core. It has been established that paper and nanocomposite produced from HCNF exhibit better mechanical properties than those produced from pure cellulose nanofibril (PCNF). Due to strong hydrophilicity of hemicellulose in the HCNF, it can act as an adhesive between the CNF and hydrophilic polymers and can improve the strength of the composites. Therefore, it has been observed to have high potential as a reinforcing filler for hydrophilic polymers [22,26,27,28]. Furthermore, it is expected that the variation in the chemical composition of the CNF may increase the utilization of wet-spun filament prepared from the CNF, based on the application purpose. 

In this study, CNFs with different chemical compositions were prepared using WDM and HPH, and were subsequently used for preparing wet-spun filaments. The characteristics of CNFs, with different chemical compositions, and the effect of chemical compositions and spinning conditions on morphological characteristics, tensile properties, and orientation of the wet-spun filaments was investigated.

## 2. Materials and Methods

Yellow poplar (*Liriodendron tulipifera* L.) was obtained from the Experimental Forest of Kangwon National University, Chuncheon, Republic of Korea. Degreased wood powder was prepared using an ethanol/benzene (1/2, *v*/*v*) solution in a soxhlet extractor, operating at 90 °C for 6 h. Sodium chlorite, acetic acid, 30% hydrogen peroxide solution, 50% sodium hydroxide solution, 45% potassium hydroxide solution, tert-butanol, and sulfuric acid were purchased from Daejung Chemical & Materials (Siheung, Republic of Korea) and were used without further purification. 

Alkaline-peroxide treatment was conducted as the pretreatment for LCNF preparation. Wood powder (5 g) was first suspended in a 0.4% sodium hydroxide solution (95 mL) and subsequently stirred at 160 rpm at 60 °C for 1 h to the dilute alkaline pre-treatment. The insoluble residue was then separated from the filtrate through vacuum filtration and was later washed with distilled water. The alkaline pretreated sample (10 g) was added to a 12% hydrogen peroxide solution (490 mL), and the pH of the suspension was then adjusted to 11.5 using a 50% sodium hydroxide solution. The treatment was conducted at 80 °C for 1 h. The residue was subsequently washed with distilled water through vacuum filtration until the pH of the filtrate became neutral. Delignification was performed from lignocellulose to prepare holocellulose based on the following sodium chlorite-acetic acid method. First, wood powder (10 g) was suspended in distilled water (600 mL), and kept at 80 °C, while being stirred at 150 rpm. The delignification reaction was then initiated by adding sodium chlorite (8 g) and acetic acid (1600 µL) into the suspension and was later continuously stirred for 1 h. Next, the same amount of sodium chlorite and acetic acid was added every hour, and the process was repeated seven times. The residue, holocellulose, was then vacuum-filtrated and washed with distilled water several times until the pH became neutral. To remove the hemicellulose in the holocellulose, alkaline treatment was conducted. The holocellulose (1 g) was soaked in 5.0% potassium hydroxide solution (25 mL) and stirred at 150 rpm at 25 ± 3 °C for 24 h. Next, to leach the hemicellulose, the reactant was kept in a bath at 80 °C and stirred for 2 h. The obtained residue, nearly purified cellulose, was neutralized with a 10% acetic acid solution and washed with excess distilled water.

Lignin content was determined using the Klason method. First, wood powder (1 g) was added to a 72% sulfuric acid solution (15 mL) and stirred for 2 h at 20 ± 3 °C. Distilled water (560 mL) was subsequently added to the mixture to dilute the sulfuric acid to 3%, and the residue was then hydrolyzed using an autoclave at 120 °C for 1 h. Next, the acid-insoluble residue was separated through vacuum filtration and later washed with excess distilled water until the wash filtrate reached a neutral pH. Subsequently, the lignin content was calculated by comparing the weight of the Klason lignin and the weight of the wood powder. The contents of hemicellulose and nearly purified cellulose were measured using the following method. First, holocellulose (10 g) was poured in a 17.5% sodium hydroxide solution (250 mL), and a reaction was subsequently performed for 30 min, while being stirred at 150 rpm at 25 ± 3 °C. After the reaction time, 10% acetic acid (250 mL) was added into the solution for neutralization. Next, a reactant was vacuum-filtrated and washed with distilled water several times. Next, the contents of hemicellulose and cellulose were calculated by subtracting the obtained weight of cellulose from the weight of holocellulose. 

The pretreated samples were suspended at 1.0 wt.% concentration and subsequently subjected to WDM (Supermasscolloider, MKCA6-2, Masuko Sangyo Co. Ltd., Kawaguchi, Japan). The rotational speed was set as 1800 rpm and the clearance between the upper and lower disks was reduced to 80–150 um from zero, at which the disks would have begun to rub. The operation was repeated until the fifth pass. The obtained CNF suspensions, after WDM, were diluted to 0.5 wt.%, and subsequently defibrillated using an HPH system (MN400BF, Micronox Co. Ltd., Sungnam, Republic of Korea) at a pressure of 30,000 psi. The defibrillation operation was repeated until fifth pass. 

The CNF suspensions prepared using WDM and HPH were centrifuged at 60,000× *g* for 15 min, and the concentrations of the precipitates were diluted to 2.5 and 3.5 wt.% by adding distilled water. Subsequently, the CNF suspension was put into in a syringe with a needle of 24G (outer diameter: 0.56 mm; inner diameter: 0.30 mm), and then placed in a syringe pump. The CNF suspensions were wet-spun in tert-butanol, with a spinning rate of 5, 10, and 20 mL/min. The wet-spun filaments were taken out from tert-butanol, and immersed in acetone for 1h to maintain linear filament during drying process. After taking it out from acetone, the filaments were hung and air-dried at 25 ± 3 °C for 2 h. 

The CNF samples for the morphology observation were prepared using the following method. The CNF suspensions were first diluted to 0.001 wt.%, and then dispersed using an ultrasonicator (VCX130PB, Sonics & Materials, Inc., Newtown, USA) for 90 s. Next, the suspensions were vacuum-filtrated on a polytetrafluoroethylene (PTFE) membrane filter in order to minimize amount of water, and at this stage, the filtration time was recorded. Next, the filtrated CNF was immersed and kept in tert-butyl alcohol for 30 min for solvent exchange. This solvent exchange was repeated five times to replace water with tert-butanol. Later, the solvent-exchanged CNFs were freeze-dried using a freeze dryer (FDB-5502, Operon Co. Ltd., Gimpo, Republic of Korea) at −55 °C for 3 h to prevent the aggregation of the CNFs. Subsequently, the freeze-dried CNFs and wet-spun filaments were coated with iridium using a high-vacuum sputter coater (EM ACE600, Leica Microsystems, Ltd., Wetzlar, Germany). Morphologies of the CNFs and filaments were observed using a scanning electron microscope (S-4800, Hitachi, Tokyo, Japan) at Central Laboratory of the Kangwon National University, Chuncheon, Republic of Korea. The diameters of CNFs and filaments were measured over 500 times and 30 times, respectively.

The filaments were kept in a thermo-hygrostat at a relative humidity of 65% to minimize the influence of variation in relative humidity on the tensile properties. The tensile properties were measured with a road cell of 5 N at a cross-head speed of 3 mm/min, with a span length of 10 mm. Ten specimens of each sample were tested and the average values were reported.

Orientation of the wet-spun filament was analyzed using a two-dimensional X-ray diffractometer (2D XRD) (Bruker D8 Discover with Vantec 500 detector) at the Korea Basic Science Institute, Daegu Center, Daegu, Republic of Korea. CuKα radiation source at 40 kV and 40 mA was used and 2D XRD analysis was carried out with a beam diameter of 1.0 mm in the transmission mode, 70 mm from the detector. A total of 40 filaments were bundled together to obtain sufficient intensity. From 2D XRD data, orientation index (α) of the CNFs in the wet-spun filament was calculated using the following equation (1) by azimuthal breath analysis.
α = (180 − βc)/180(1)
where βc is the half width of the azimuthal direction of the equatorial reflection of (200) the plane obtained from the 2D XRD patterns.

## 3. Results and Discussion

### 3.1. Chemical Composition

The chemical composition of the CNFs was analyzed, as shown in Table 1. The LCNF consisted of 79.1% cellulose, 6.0% hemicellulose, and 13.0% Klason lignin. The HCNF and nearly purified-cellulose nanofibril (NP-CNF) did not have any lignin content, while the hemicellulose contents were found to be 36.6% and 5.1%, respectively. The LCNF was observed to have a significantly lower content of hemicellulose than HCNF. During the delignification process by alkaline-peroxide treatment, the pH of the reactant was kept at 11.5. The alkaline condition may have partially removed the hemicellulose in the LCNF. In the NP-CNF sample, the hemicellulose remained at 5%. In a previous study [21], PCNF was prepared without lignin and hemicellulose through a strong alkaline treatment using holocellulose and a 17.5% sodium hydroxide solvent. However, it is known that under a strong alkaline condition, crystals in cellulose get transferred from cellulose type I with parallel structure to cellulose type II with anti-parallel structure. To maintain the cellulose I structure, weak alkaline treatment, using a 5% KOH solvent, was conducted, and the NP-CNF, with 5% hemicellulose, was obtained. The crystal structure of the samples is described below. 

### 3.2. Morphological Characteristics

Morphological characteristic of LCNF, HCNF, and NP-CNF are shown in Figure 1. In the LCNF sample, fibers with the diameters of 18–25 nm were observed, alongside thick fibers with the diameters of approximately 50 nm. The HCNF and NP-CNF samples displayed a finer and more uniform morphology, with a diameter of approximately 18 nm. The average diameters, specific surface areas, and filtration times of LCNF, HCNF, and NP-CNF are summarized in Table 1. The LCNF was observed to have an average diameter of 25.9 nm, while the HCNF and NP-CNF were observed to have similar average diameters of approximately 18.0–18.5 nm. The specific surface areas of LCNF, HCNF, and NP-CNF were found to be 142.8, 182.1, and 180.5 m^2^/g, respectively (Table 1), which might be because a reduction in the average diameter of the CNF increased the specific surface area. In a previous study [21], the effect of chemical composition on the defibrillation efficiency of the CNFs was investigated. Results indicated that the existence of lignin interrupted the defibrillation of cellulose. Thus, the removal of lignin improved the defibrillation efficiency, indicating the decrease in the average diameter of the CNF. The filtration time of the CNF suspensions can be used as a criterion to indirectly evaluate the surface characteristic of the CNFs [21,22]. The filtration time of the CNF suspensions decreased in the order of, LCNF < NP-CNF < HCNF (Table 1). The LCNF sample was observed to contain hydrophobic lignin and had a smaller specific surface area than the other samples. This might have resulted in a shorter filtration time of the LCNF suspension [21]. Although HCNF and NP-CNF were observed to have similar average diameters and specific surface areas, the HCNF suspension was observed to have a longer filtration time. This might be because the hygroscopic hemicellulose in HCNF had a strong hydrophilicity, which increased the water holding capacity [22,26]. 

Figure 2 indicates the morphological characteristics of the wet-spun filaments prepared from 3.5 wt.% LCNF, HCNF, and NP-CNF at a spinning rate of 20 mL/min. In all the samples, aggregated CNFs were observed on the wet-spun filaments. The wet-spun LCNF filament was observed to have a thicker diameter with a rougher surface on the filament, while the wet-spun HCNF filament was observed to have a smaller diameter with a smoother surface than the other samples. The average diameters and densities of the wet-spun filaments under different spinning conditions are summarized in Table 2. The wet-spun filaments from 3.5 wt.% LCNF, HCNF, and NP-CNF suspensions were observed to have diameters of 181, 199, and 204 nm, respectively. The density was observed to decrease in the order of HCNF > NP-CNF > LCNF. The relationship between the average diameter and density can be probably explained by the aggregation of CNFs due to its hydrophilic property [21]. As the hydrophilicity of the CNFs increased in the order of LCNF < NP-CNF < HCNF, the hydrogen bonding between the CNFs was observed to increase, which might have increased the aggregation between the CNFs due to the shrinkage phenomenon. Thus, the wet-spun filament, with higher hydrophilicity, may have a narrower diameter and higher density. Park et al. (2017) [21] prepared the LCNF, HCNF, and PCNF through wet-disk milling using six different wood species and fabricated paper sheets from LCNF, HCNF, and PCNF. The density of the sheets fabricated from LCNF, HCNF, and PCNF was observed to be 0.33–0.88, 0.96–1.63, and 1.04–1.31 g/cm^3^, respectively. It was also demonstrated that the hydrogen bonding due to hydrophilicity affected the density of the CNF products. In the wet-spun HCNF filaments, as the concentration decreased from 3.5 to 2.5 wt.%, the average diameter was observed to decrease. Moreover, as the spinning rate was increased from 5 to 20 mL/min, the average diameter of the wet-spun HCNF filament was observed to decrease.

### 3.3. X-Ray Diffraction Anlaysis

Figure 3 shows the 2D X-ray diffractograms of the wet-spun filaments prepared from 3.5 wt.% concentration of LCNF, HCNF, and NP-CNF at a spinning rate of 20 mL/min. According to the 2D X-ray diffractograms, the wet-spun filaments displayed a typical peak for cellulose I, corresponding to the (110) and (200) planes [4,22]. In general, a randomly fabricated cellulose film has shown ring patterns along the azimuthal angle in a 2D X-ray diffractogram [4,18]. In this study, the reflections were observed to be strong at the middle azimuth, with two arc patterns. These arc patterns, as shown in Figure 3, indicate that the CNFs oriented in the wet-spun filaments. 

To investigate the orientation of the wet-spun filaments, with different concentrations and spinning rates, the relative intensity of the (200) plane along the azimuthal was measured using the 2D XRD diffractograms, as shown in Figure 4. The orientation index was calculated from the relative intensity of the (200) plane, as shown in Figure 4 and summarized in Table 2. The orientation indexes of the wet-spun filaments produced from 3.5 wt.% LCNF, HCNF, and NP-CNF were found to be 0.57, 0.60, and 0.60, respectively. The orientation indexes of the wet-spun filaments produced from HCNF and NP-CNF were observed to be similar, while the wet-spun filament produced from LCNF was observed to have a lower orientation index than the other samples. Furthermore, the LCNF sample was observed to have a thicker diameter than the other samples, which may have inhibited the orientation in the wet-spun filaments. The effect of CNF diameter on the orientation can be inferred from Lundahl et al. (2016) [18]. Lundahl et al. (2016) [18] prepared a filament via the wet-spinning from a pristine CNF and TOCNF. The diameters of the CNF and TOCNF were measured from the profile of an atomic force microscope and were found to be approximately 4–5 and 2 nm, respectively. Comparison of the azimuthal integration in the 2D XRD profiles on the (200) plane indicated that the wet-spun filament produced from TOCNF exhibited higher orientation along the filament axis than that produced from CNF. In the wet-spun HCNF filaments, it was found that the orientation index increased from 5 to 20 mL/min as the spinning rate was increased. This indicated that the increase in the spinning rate enhanced the shearing force on the CNFs and, consequently, promoted the orientation of the CNF [17,18]. Furthermore, at the same spinning rate of 20 mL/min, as the concentration of the HCNF suspension decreased from 3.5 to 2.5 wt.%, the orientation index of the wet-spun HCNF filaments was observed to increase. This could be because at a higher concentration, frequent contact between the CNF fibers interfered with the orientation of the CNF fibers, induced by the shear-force generated by the flow of the fluid [18]. 

### 3.4. Tensile Properties

Figure 5 shows the tensile strengths, specific tensile strengths, elastic modulus, and elongations at break of the wet-spun filaments made from LCNF, HCNF, and NP-CNF under different spinning conditions. Under the same spinning condition, the tensile strength of the wet-spun LCNF filament was observed to be lower than that of the other samples, which might be because of the existence of hydrophobic lignin in the fiber. The hydrophobic property of the lignin is known to weaken the hydrogen bonding that can increase the tensile strength between the CNFs [7,21]. Consequently, the tensile strength can be reduced [21]. In addition, the LCNF was observed to have a smaller specific surface area than the HCNF and NP-CNF. The wet-spun HCNF filament was observed to have a higher tensile strength among all the samples. The HCNF contains hemicellulose without lignin and a core-shell structure with a cellulose core and a shell of hemicellulose [26,27]. As the hemicellulose in the HCNF has extremely high hydrophilicity, it can significantly enhance the adhesion between the CNFs [22,26], thereby improving the tensile strength of the wet-spun filament. In all the samples, it was observed that as the concentration decreased and the spinning rate increased, the tensile strength of the wet-spun filaments increased. This can be explained by the orientation of the CNFs in the wet-spun filament. The orientation of the CNF is an important factor affecting the tensile strength of the wet-spun filaments. According to Table 2, the orientation index of the wet-spun filament was observed to increase with increasing spinning rate and decreasing concentration. The crystals in the CNF have been observed to have an aspect ratio and extremely high strength and elastic modulus in the longitudinal direction [4]. It can be observed from Figure 5 that the elastic modulus also increased with decreasing CNF concentration and increasing spinning rate. This was also due to the increase in the orientation index. Thus, the increase in the CNF alignment along the wet-spun filament axis has been considered to improve the tensile properties [4,17,18]. The tensile strength was also observed to be affected by the density of the wet-spun filaments. Additionally, the specific tensile strength was calculated, as shown in Figure 5. As the difference in the densities between the wet-spun filaments produced from LCNF, HCNF, and NP-CNF was not significant, the specific tensile strength was observed to exhibit a similar trend as that of the tensile strength. Furthermore, the elongation at break of the wet-spun LCNF filament was observed to be slightly lower than that of the other samples, and the wet-spun filament produced from HCNF and NP-CNF were observed to exhibit similar elongation at break. 

## 4. Conclusions

CNFs with different chemical compositions, LCNF, HCNF, and NP-CNF, were prepared in this study through wet-disk milling and high-pressure homogenization. Filaments were prepared from LCNF, HCNF, and NP-CNF through wet-spinning. The LCNF was observed to have a thicker diameter than that of the HCNF and NP-CNF, indicating a smaller specific surface area. The filtration time for CNF suspension was longer and followed the order of LCNF < NP-CNF < HCNF. The shortest and longest filtration times for LCNF and HCNF were caused by lignin with hydrophobicity and the water holding capacity of hygroscopic hemicellulose, respectively. The wet-spun filament produced from LCNF, HCNF, and NP-CNF was observed to exhibit a typical cellulose type I. An increase in the spinning rate and decrease in the CNF concentration were found to increase the orientation index. This indicated that the increased shearing force due to a high spinning rate promoted the orientation of the CNFs in the wet-spun filaments. The wet-spun HCNF filament was observed to have a higher tensile strength, and the tensile strength of the wet-spun LCNF filament was found to be lower than that of the other samples. The high content of hemicellulose was found to improve the tensile strength due to its strong hydrophilicity. In each sample, as the concentration of the CNF suspension decreased and the spinning rate increased, the tensile strength and elastic modulus increased. This was because an increase in the orientation due to a lower concentration and high spinning rate improved the tensile strength and elastic modulus of the wet-spun filaments. 

This study has demonstrated that the properties of the wet-spun filaments can be varied, depending upon the chemical composition of the CNFs. It was also found that an increase in the CNF orientation in the wet-spun filament resulted in the wet-spun filament having excellent mechanical properties. The difference in the properties, dependent upon the chemical composition of the CNF, can provide important information on how and where the CNF can be applied. 

## Figures and Tables

**Figure 1 polymers-12-00949-f001:**
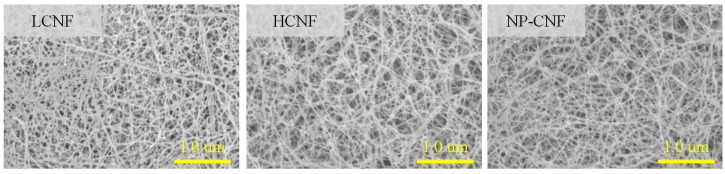
Morphological characteristics of LCNF, HCNF, NP-CNF.

**Figure 2 polymers-12-00949-f002:**
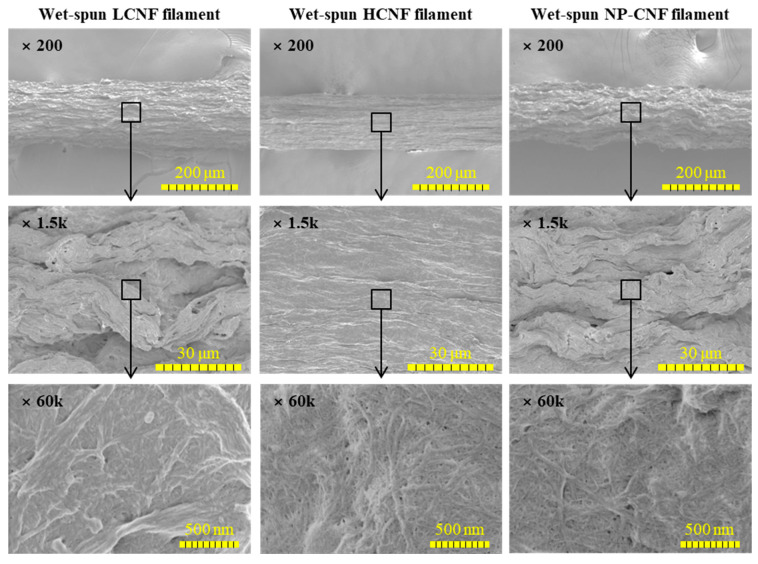
Morphological characteristics of the wet-spun filament prepared from 3.5 wt.% concentration of LCNF, HCNF, and NP-CNF at a spinning rate of 20 mL/min.

**Figure 3 polymers-12-00949-f003:**
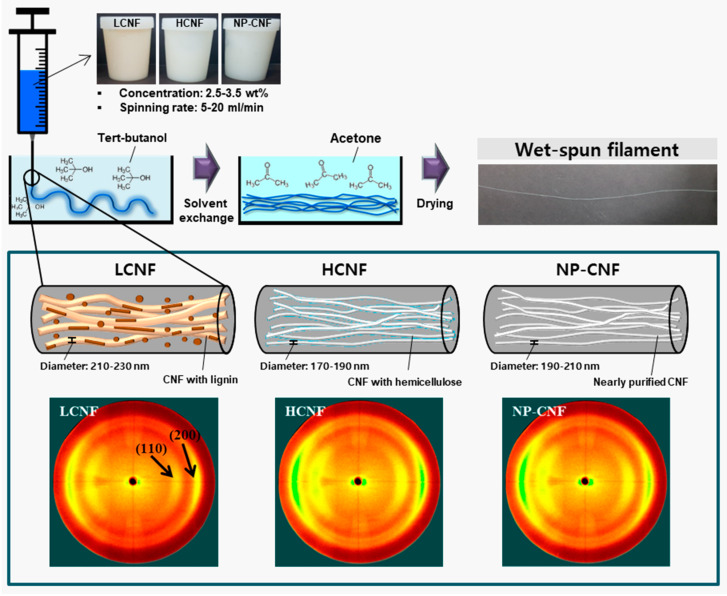
Schematic diagram of wet-spinning and 2D X-ray diffractograms of the wet-spun filament filaments made from 3.5 wt.% concentration of LCNF, HCNF, and NP-CNF at a spinning rate of 20 mL/min.

**Figure 4 polymers-12-00949-f004:**
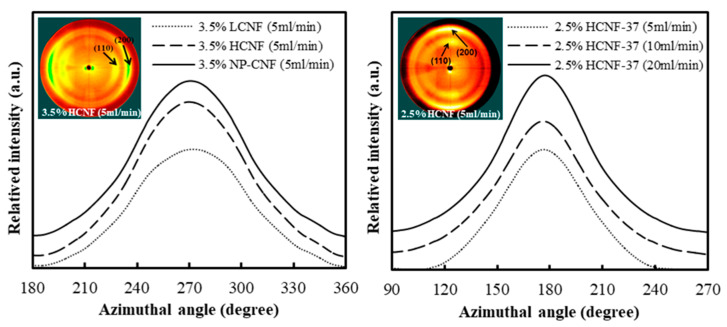
Azimuthal profiles of the (200) reflections from 2D X-ray diffractograms of the wet-spun filaments made from LCNF, HCNF, and NP-CNF, with different concentrations and spinning rates.

**Figure 5 polymers-12-00949-f005:**
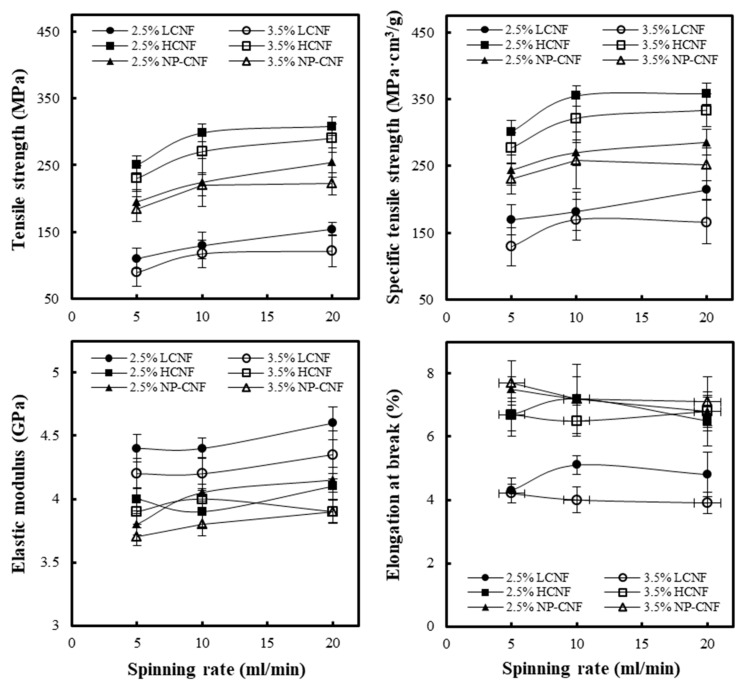
Tensile strength, specific tensile strength, elastic modulus, and elongation at the break of the wet-spun filaments, made of LCNF, HCNF, and NP-CNF, at different concentrations of CNFs and spinning rates.

**Table 1 polymers-12-00949-t001:** Chemical composition, average diameter, specific surface area, and filtration time of lignocellulose nanofibril (LCNF), holocellulose nanofibril (HCNF), and nearly purified-cellulose nanofibril (NP-CNF).

	Chemical Composition (%)			Number of HPH Passes	Average Diameter (nm)	Specific Surface Area (m^2^/g)	Filtration Time (sec)
	Cellulose	Hemicellulose	Klason Lignin				
LCNF	79.1	6.0	13.0	5	25.9 ± 11.4	142.8	259
HCNF	63.3	36.6	-	5	18.5 ± 3.1	182.1	394
NP-CNF	94.9	5.1	-	5	18.3 ± 2.4	180.5	321

**Table 2 polymers-12-00949-t002:** Average diameters and orientation indexes of the wet-spun filaments with different concentration of CNFs and spinning rates.

	Concentration (wt.%)	Spinning Rate (mL/min)	Average Diameter (nm)	Density (g/cm^3^)	Orientation Index
LCNF	3.5	20	224 ± 15	0.73 ± 0.02	0.57
HCNF	2.5	5	175 ± 17	0.83 ± 0.04	0.64
		10	151 ± 11	0.84 ± 0.03	0.66
		20	143 ± 9	0.86 ± 0.01	0.67
	3.5	20	181 ± 13	0.87 ± 0.05	0.60
NP-CNF	3.5	20	199 ± 16	0.83 ± 0.03	0.60

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
