# Peer review of "Preparation and Characteristics of Wet-Spun Filament Made of Cellulose Nanofibrils with Different Chemical Compositions"

_polymers, 2020, doi:10.3390/polym12040949_

Round 1

Reviewer 1 Report

In this research, the authors reported the preparation of filamentous fibers out of cellulose nanofibrils with different chemical compositions and their physical and mechanical properties based on geometrical analysis. Considering that this fundamental study on regenerated cellulosic fibers may contribute to the guided development of high value-added cellulose products, this research is informative to researchers in the related fields. Nonetheless, there are several issues in this manuscript which should be properly handled before this manuscript is considered for the publication in Polymers.

  1. Although there exist several similar fundamental studies in the literature, the originality/novelty of the present research should be clarified in the title, abstract, introduction, and, possibly, experimental, if any.
    [M. Sain et al., J. Reinf. Plast. Comp., 2015, 24, 1259]
    [A. Abdulkhani, et al., Polym. Test., 2014, 35, 73]

  1. It is highly recommended that the expression of “filtration time” should be rephrased. Although I suppose that authors meant that the CNF/CNC dispersion was filtered for solvent exchange, it can mislead readers by the preparation of cellulosic fibril mesh as filter for purification.

  1. It is not convincing that it takes less time to remove water through the (hydrophobic) LCNF than the (relatively hydrophilic) HCNF or NP-CNF, considering that water penetrate though the hydrophobic (e.g., PTFE) membrane typically longer time than the hydrophilic membrane (e.g., cellulose acetate membrane). Authors should doublecheck the experimental condition and explain more clearly about the apparent filtration time.

  1. The experimental explanation and discussion about the upper right image in Figure 3 is not provided in the manuscript. What is the main purpose of experiments and how were they conducted? Why were the different metrics of mass employed?

  1. In polymer physics, the response of mechanical properties to thermal and rheological stimuli is very informative. It is highly recommended that authors should add the thermal and rheological characterizations such as DSC, TGA, DMA, and the corresponding discussion.

Author Response

Comment 1.

Although there exist several similar fundamental studies in the literature, the originality/novelty of the present research should be clarified in the title, abstract, introduction, and, possibly, experimental, if any.

[M. Sain et al., J. Reinf. Plast. Comp., 2015, 24, 1259]

[A. Abdulkhani, et al., Polym. Test., 2014, 35, 73]

Response 1.

Thank you for your review. The nanocellulose has been attracting an interest as a reinforcing filler for composite materials. However, until now the application to the textile industry using only nanocellulose has been limited despite of its outstanding properties. In this study, as an effort to utilize nanocellulose in the textile industry, we’ve prepared the filament  from CNFs via wet-spinning. Our research team has studied the variation in properties of CNF based products dependent on chemical composition in CNFs. We’ve expected that the properties of the wet-spun filaments can be varied with changing chemical composition, and investigated its effect on physico-mechanical properties of the wet-spun filament. Therefore, the chemical composition in CNF is mentioned in the title and introduction. Expected effects resulting from varying chemical composition in CNFs are also described in the Introduction. And the results for that on properties of filaments were demonstrated in Results. Thank you.

Comment 2.

It is highly recommended that the expression of “filtration time” should be rephrased. Although I suppose that authors meant that the CNF/CNC dispersion was filtered for solvent exchange, it can mislead readers by the preparation of cellulosic fibril mesh as filter for purification.

Response 2.

Thank you for your valuable comment. We’ve modified the experiment part clearly for readers to understand this method clearly. Thank you.

Comment 3.

It is not convincing that it takes less time to remove water through the (hydrophobic) LCNF than the (relatively hydrophilic) HCNF or NP-CNF, considering that water penetrate though the hydrophobic (e.g., PTFE) membrane typically longer time than the hydrophilic membrane (e.g., cellulose acetate membrane). Authors should doublecheck the experimental condition and explain more clearly about the apparent filtration time.

Response 3.

We agree that your opinion that filtration time can be varied dependent on hydrophobic or hydrophilic membrane filters. The filtration time is kind of indirect way to measure water retention of CNFs. However, although LCNF contains hydrophobic lignin, even water in LCNF suspension is not hydrophobic. The existence of lignin in LCNF only affects the decrease of the ability to water retention. Thus, in both hydrophilic and hydrophobic filters, the filtration time of LCNF tend to be shorter than HCNF and NP-CNF.

Comment 4.

The experimental explanation and discussion about the upper right image in Figure 3 is not provided in the manuscript. What is the main purpose of experiments and how were they conducted? Why were the different metrics of mass employed?

Response 4.

Thank you for your critical comment. The upper right images in Figure 3 was used to easily express the difference strength resulting from chemical composition. However, since the tensile strength and elastic modulus were already in figure 5, we eliminated the upper right images in Figure 3. Thank you.

Comment 5.

In polymer physics, the response of mechanical properties to thermal and rheological stimuli is very informative. It is highly recommended that authors should add the thermal and rheological characterizations such as DSC, TGA, DMA, and the corresponding discussion.

Response 5.

Yes, I agree that mechanical properties of polymers will be sensitive with thermal and rheological stimuli. The experiments to figure out the dependence of other rheological and thermal properties of CNFs with different concentrations using DSC, TGA, DMA on the mechanical properties would be better for another research topic.  Once again, thanks for your valuable recommendation.

Reviewer 2 Report

This is an interesting paper which combines the characterisation of starting material with manufacturing, characterisation of the wet-spun filament, and evaluation of the mechanical properties.  As one might expect in a paper which covers such a lot of ground some area are better described and discussed than others but overall I believe it is a useful piece of work that is appropriate for publication in the journal Polymers.  There are a few issues which I would like to see addressed as detailed below:

  1. The phrase “filament fibre” is used extensively throughout the paper. This is tautology and although the two words can have very specific meanings, in this case they mean the same.
  2. There are many acronyms in the paper and I found myself constantly ging backwards and forwards as I read the paper to seek clarification. A list prior to the introduction would be very helpful to the reader.
  3. Page 2 line 56: The diameter of the CNF is very helpfully provided but what of the length range?
  4. P 3: The description of the preparation of the starting material for the filament spinning process is described in great detail, but the spinning process itself gets barely five lines.  This needs to be expanded, perhaps with a better schematic than provided in Figure 3.
  5. P3 l 139: Define CNC.
  6. P4 Table 1: The accuracy of the composition and diameter seems rather optimistic for a natural material.  These values are clearly mean values but how were they determined from a statistical standpoint?  I am particularly concerned with the diameter which will of course vary along fibres.
  7. P5 Figure 1: Scale bars in yellow are not ideal and the magnification chosen does not allow one to appreciate the individual fibrils.  Against a diameter of around 20 nm I would expect to see some higher magnification images of these fibrils.
  8. P6 Figure 2: In a similar vein it would be helpful to show intermediate magnification SEM images, say x5k.
  9. P7 l 251: Whilst I can just about live with diffractogram (although it is a very old-fashioned term) spectrogram is definitely wrong, both as a word and in context!
  10. P8 Figure 4: Not really sure that this adds anything to the story given that it yields a set of very similar data and is used to calculate the Orientation Index of Table2.  What samples do the two diffraction patterns included in Figure 4 refer to?

None of these issues are show-stoppers but attention to these pints will make for a significantly better paper.

Author Response

Comment 1.

The phrase “filament fibre” is used extensively throughout the paper. This is tautology and although the two words can have very specific meanings, in this case they mean the same.

Response 1.

Thank you for your valuable comment. In order to avoid duplication of the meaning, we’ve modified the “filament fibre” to “filament”. Thank you.

Comment 2.

There are many acronyms in the paper and I found myself constantly going backwards and forwards as I read the paper to seek clarification. A list prior to the introduction would be very helpful to the reader.

Response 2.

We’ve used the abbreviations to indicate lignocellulose nanofibril (LCNF), holocellulose nanofibril (HCNF), and nearly purified nanofibril (NP-CNF). The acronyms have already been introduced in ABSTRACT as well as INTRODUCTION. We think it will be easy for readers to understand the meaning of the abbreviations. Thank you.

Comment 3.

Page 2 line 56: The diameter of the CNF is very helpfully provided but what of the length range?

Response 3.

Thank you for your critical comment. The diameter of CNF is nano-scale and can be determined through microscopes. However, the CNF has a length with several micrometers and tangled. It is very difficult to measure the length through a microscope. Thus, many researches have tried to develop the ways to predict the length.

Comment 4.

P 3: The description of the preparation of the starting material for the filament spinning process is described in great detail, but the spinning process itself gets barely five lines.  This needs to be expanded, perhaps with a better schematic than provided in Figure 3.

Response 4.

Thank you for your critical comment. We’ve modified the experimental methods for spinning in more detail. Thank you.

Comment 5.

P3 l 139: Define CNC.

Response 5.

it was mistyped. We’ve modified it to “CNF” and checked all word in this manuscript meticulously.

Comment 6.

P4 Table 1: The accuracy of the composition and diameter seems rather optimistic for a natural material.  These values are clearly mean values but how were they determined from a statistical standpoint?  I am particularly concerned with the diameter which will of course vary along fibres.

Response 6.

To determine weight percentages of cellulose, hemicellulose, and cellulose we repeated the experiment at least thrice. And the diameter of filament was measure more than 30 times. To give credibility to the figures in the tables, we’ve specified the number of repeat measurements for diameter and chemical composition in Experimental part.

Comment 7.

P5 Figure 1: Scale bars in yellow are not ideal and the magnification chosen does not allow one to appreciate the individual fibrils.  Against a diameter of around 20 nm I would expect to see some higher magnification images of these fibrils.

Response 7.

We agree with your opinion. However, because the wet spun fiber we made is not a flat material on the nano or micron-scale, the increase in magnification over 60K for the fiber is technically very difficult, that causes charging phenomenon and rapid resolution reduction. We’ve just divided the scale bar into ten equal parts to further refine the scale bar. Thank you.

Comment 8.

P6 Figure 2: In a similar vein it would be helpful to show intermediate magnification SEM images, say x5k.

Response 8.

Thank you for your review. We’ve added the new SEM images for intermediate magnification. Thank you.

Comment 9.

P7 l 251: Whilst I can just about live with diffractogram (although it is a very old-fashioned term) spectrogram is definitely wrong, both as a word and in context!

Response 9.

Thank you for your review.The 2D XRD patterns being showed in this manuscript is kind of diffraction diagram from X-ray. Thus, we use the word 'diffractogram'. and we the modified the word 'spectrogram' to 'diffractogram'. Thank you.

Comment 10.

P8 Figure 4: Not really sure that this adds anything to the story given that it yields a set of very similar data and is used to calculate the Orientation Index of Table2.  What samples do the two diffraction patterns included in Figure 4 refer to?

Response 10.

In table 2, orientation indexes with different types of CNFs and spinning rates were indicated. As you know, the orientation indexes were calculated from Figure 4. In figure 4, due to the difference in the initial setting value, the maximum peak of the azimuth is shown at 0° and 180° degrees in the graph on the left, but the maximum peak at 90° and 270° is shown in the graph on the right. Although this difference does not affect the orientation index, the some reader may misunderstand the azimuth value, so an example photo is placed at the top left of each graph. We agree that the example photos need to indicate what the result of the sample is. Thus, we added the type of sample for the indicated 2D XRD images. Thank you.

Round 2

Reviewer 1 Report

The authors properly addressed most of reviewers' comments and revised the manuscript accordingly. In my opinion, this work is ready to be published as it is. 

Reviewer 2 Report

All comments attended to concisely.